# Pathways from Childhood Economic Conditions to Adult Mortality in a 1953 Stockholm Cohort: The Intermediate Role of Personal Attributes and Socioeconomic Career

**DOI:** 10.3390/ijerph19127279

**Published:** 2022-06-14

**Authors:** Klara Gurzo, Bitte Modin, Pekka Martikainen, Olof Östergren

**Affiliations:** 1Department of Public Health Sciences, Stockholm University, SE-106 91 Stockholm, Sweden; bitte.modin@su.se (B.M.); pekka.martikainen@helsinki.fi (P.M.); olof.ostergren@su.se (O.Ö.); 2Max Planck Institute for Demographic Research, Konrad-Zuse-Str., 18057 Rostock, Germany; 3Population Research Unit, University of Helsinki, 00014 Helsinki, Finland; 4Aging Research Center, Karolinska Institutet, Tomtebodavägen 18A, SE-171 65 Solna, Sweden

**Keywords:** childhood conditions, parental income, cognitive ability, adult income, all-cause mortality

## Abstract

Although both childhood and adult economic conditions have been found to be associated with mortality, independently or in combination with each other, less is known about the role of intermediate factors between these two life stages. This study explores the pathways between childhood economic conditions and adult mortality by taking personal attributes as well as adult socioeconomic career into consideration. Further, we investigate the role of intergenerational income mobility for adult mortality. We used data from a prospective cohort study of individuals that were born in 1953 and residing in Stockholm, Sweden, in 1963 who were followed for mortality between 2002 and 2021 (n = 11,325). We fit Cox proportional hazards models to assess the association of parental income, cognitive ability, social skills, educational attainment, occupational status, and adult income with mortality. The income mobility is operationalized as the interaction between parental and adult income. Our results show that the association between parental income and adult mortality is modest and largely operates through cognitive ability and adult educational attainment. However, our results do not provide support for there being an effect of intergenerational income mobility on adult mortality. In a Swedish cohort who grew up in a comparatively egalitarian society during the 1950s and 1960s, childhood economic conditions were found to play a distinct but relatively small role for later mortality.

## 1. Introduction

The importance of childhood socioeconomic conditions for adult health and mortality has been the topic of extensive research within the fields of social science and public health. Register-based studies have typically found that exposure to various aspects of poor socioeconomic circumstances during childhood are associated with a higher risk of mortality in adulthood [1,2,3]. However, studies that are based on survey data on childhood socioeconomic conditions show more inconsistent patterns, possibly because of recall biases in retrospective reports or non-participation [3,4,5].

While the associations of parents’ social class and educational level with children’s later mortality risk have been extensively studied [1,6,7,8,9], fewer studies have investigated the long-term association between childhood economic conditions and adult mortality risk. One study of a Swedish cohort that was born in 1928 demonstrated a negative association between parental income and later-life mortality [10]. Moreover, a recent study on Danish individuals that were born between 1980 and 1998 found that cumulative material deprivation during childhood (in terms of family poverty and long-term parental unemployment) was associated with a higher mortality risk in early adult life compared to children with few or no adversities [11].

The fact that people’s socioeconomic careers are dependent on the social positions of their parents raises the question of whether economic conditions in childhood have an independent association with mortality once adult socioeconomic positions and personal attributes have been taken into consideration. Accordingly, the pathway model posits that childhood socioeconomic position influences adult health through the intergenerational transmission of socioeconomic position, with little or no direct effect on adult health [3,9]. In line with this, several studies have found that adjusting for adult socioeconomic position substantially attenuates the association between childhood socioeconomic circumstances and adult mortality [5,7,9,12,13], sometimes to the extent that the association completely disappears [5,9,13]. However, it is reasonable to assume that the relative importance of childhood versus adult socioeconomic conditions for adult health depends on the specific health outcome as well as time and place [3,4,14].

How well individuals succeed in their socioeconomic careers also depends on personal attributes. Examples of attributes that may be of particular importance here are cognitive ability and social skills. These attributes are part of the individual’s potential of acquiring a later-life social position and the income that comes with it [15,16,17,18]. At the same time, cognitive ability and social skills are influenced by childhood socioeconomic conditions [19] and may be intermediate factors in the association between childhood conditions and adult socioeconomic achievement and mortality. Prior studies have shown that differences in cognitive ability and social skills partly explain the association between adult socioeconomic position and health [20], but that they also are associated with mortality independently of both childhood and adult socioeconomic position [21,22,23].

Related to the above is also the question of whether the mobility from childhood to adult socioeconomic position plays a role itself for an individual’s risk of disease and mortality. Social mobility refers to the change between status of origin and destination. Most studies on this topic tend to find that upwardly mobile individuals have worse health than those they join and better health than those they leave behind, while the downwardly mobile have better health than those they join and worse health than those they leave behind [6,8,24,25,26,27]. Whether these patterns can be simply attributed to the sum of the different positions that are held by an individual across the life course or if the mobility process in itself is related to health remains unclear. Some recent research has even raised the possibility of negative health effects of ‘social climbing’ by showing that upward mobility is associated with certain stress-related inflammatory markers and health problems [28,29,30].

The vast majority of studies on the health-related consequences of social mobility are based on occupational status [6,8,24,25,26,29] or educational attainment [27,28]. Yet, education, occupation, and income are independently associated with health [31,32,33], in part through different processes [34,35]. Consequently, there is a need to further explore how income and income mobility over the life course are related to later health and mortality. Awareness of the bidirectional nature of the relationship between adult income and health is crucial in such analyses. Individual health and socioeconomic careers develop together over the life course and health can both shape the individuals’ socioeconomic career and be shaped by socioeconomic position [36,37,38]. Income is dependent on the individual’s capacity to participate in the labor market, which requires a certain level of physical and mental health. Therefore, adult income is not simply a measurement of adult socioeconomic position and available material resources but can also reflect childhood and adult health.

Although an extensive literature has explored the relationship between adult mortality and socioeconomic position in childhood and adulthood, a need remains to consider the role of intermediate factors between these two life stages. Accordingly, this study aims to explore the pathways from childhood economic conditions to all-cause mortality (between 48 and 68 years of age) through cognitive ability, social skills, educational attainment, occupational class, and adult income. Moreover, we investigate if intergenerational income mobility is related to mortality.

We analyzed data on a Swedish cohort that were born in 1953 who lived in the Stockholm metropolitan area at age 10. A unique feature of our data is that they comprise not only register information on the cohort members’ later life social position but also sixth grade information on their cognitive ability and sociometric peer nominations for “best friend”. We expect that the association between childhood economic conditions and adult mortality will be attenuated when adjusting for these personal attributes. We also expect that once we adjust for adult income, the association between childhood economic conditions will decrease or vanish. Consequently, we anticipate the association between intergenerational income mobility and adult mortality to be modest as well. The study is structured as follows: first we present our data material, measures, and our analytical strategy. Then, we describe our results and present several sensitivity analyses to test the robustness of our findings. Finally, we discuss our results in light of previous research and in relation to our original research question.

## 2. Materials and Methods

### 2.1. Data

We used the Stockholm Birth Cohort Multigenerational Study (SBC Multigen). The SBC Multigen is a prospective cohort study which links survey data of children who were born in 1953 and lived in the greater Stockholm area in 1963 (N = 15,117) with register data on them and their parents at several stages across their lives [39]. The study follows the cohort until around retirement age. The Stockholm Regional Ethical Review Board provided ethical permission for compiling the data, as well as the use of the data for research purposes (reg No. 2017/684-32).

We first restricted the sample to those who were alive and living in Sweden at the beginning of 2002 (n = 13,601). We then excluded cohort members who had missing information on any of the variables that were included in the analysis. The final analytical sample consisted of 11,325 individuals, whom we followed with respect to mortality from January 2002 to the middle of May 2021 when they were between 48 and 68 years old. The descriptive statistics are presented in Table 1.

### 2.2. Measures

Information on parental income was retrieved from the Register of Population and Income from 1963 and 1970, when the cohort members were aged 10 and 17. Parental income was measured through the average pretax labor income of both the father and the mother in both years. The measures were inflation adjusted in 2001 SEK before adding them together and taking their average. We used information on both parents’ income even if the cohort member lived in a single parent household, assuming that both parents supported the child financially. Cognitive ability was measured via a cognitive test (consisting of one verbal, one spatial, and one numerical part) that was conducted in 1966 when the cohort members were aged 13 or 14 years. In the same year, the cohort members also took part in a sociometric assessment, where they were asked to nominate a maximum of three of their classmates as best friends. The number of nominations that were received by the cohort members from their classmates was used as a proxy for their social skills.

We used register information on educational attainment from 1991 and on occupational class from 1990. Educational attainment categories were pre-primary, primary, lower secondary, upper secondary, post-secondary non-tertiary, first stage of tertiary, and second stage of tertiary education. The Swedish Socioeconomic Index (SEI) was used to indicate occupational class. Occupations were categorized as low manual, high manual, low non-manual, and high non-manual. Individuals whose occupational class was unknown were included as a separate category. Information on adult income was collected from LISA (Longitudinal Integrated Database for Health Insurance and Labour Market Studies) and defined as the average of pretax labor income from 1990 to 2001. We included multiple years of income in order to reduce life cycle and attenuation bias. Adult income measures included income from work, but also social benefits, for example sickness benefits and parental allowance, but not unemployment benefits (variable name in register: forvers). Adult income was inflation adjusted in 2001 SEK.

Information on all-cause mortality (January 2002–May 2021) was extracted from the Swedish Causes of Death Register.

### 2.3. Methods

Cox proportional hazards models were used to investigate our research questions. We set the starting date to 1 January 2002 and the date of death as the time of event. Emigration or end of follow-up (13 May 2021) were set as the time of right-censoring if no event happened. First, we estimated associations between each of the included covariates and mortality adjusting only for sex. Then, we estimated a series of models adjusting for covariates in the order that they were measured or established. Global and detailed post-estimation tests (based on Schoenfeld residuals) indicated that the proportional hazards assumption was not violated.

Due to the curvilinear relationship between mortality and adult income [40], we used the log of adult income. To avoid a disproportionate influence from extreme values, we applied bottom coding at the 4th percentile and top coding at the 99th percentile of adult income. Asymmetrical top and bottom coding was applied because the same value was assigned to those individuals registered with zero income (3% of the sample) and to the first percentile of those with non-zero income. We found that compared to the log of parental income, transforming parental income into percentiles and modeling it using a linear term yielded better model fit based on Akaike’s Information Criteria (AIC) and Bayesian Information Criteria (BIC). Thus, we employed linear parental income rank and log transformed adult income.

Some cohort members had zero registered labor income during 1990 and 2001. Having no registered labor income for an extended period of time might be due to various reasons (nontaxable incomes, tax evasion, household work, inactivity in the labor market due to health problems or functional disabilities, etc.). We included a binary variable indicating whether the cohort member had zero income to account for this.

Finally, we fitted an interaction model including tertiles of parental and adult income as well as interaction terms for all the possible combinations of parental and adult income. Using quantiles, in this case tertiles, allows the functional form of the association between mortality and income to vary. Margin plots were used to visualize the association of parental and adult income tertiles with mortality. All analyses were carried out in Stata 16.

## 3. Results

The first column in Table 2 presents associations for each of the independent variables with mortality adjusted for sex. Since all the cohort members are born in the same year, results from these models are equivalent to sex and age-adjusted models. Higher parental income was associated with lower mortality in adulthood. One unit increase in parental income rank was associated with a 0.004 (*p* < 0.001) unit decrease in the log of relative hazard of mortality, which corresponds to HR = 0.996 and 95% confidence interval (CI) = 0.994–0.998. Each independent variable was significantly associated with mortality in the direction that was expected based on previous literature; higher social skills and cognitive ability were associated with lower mortality, as were higher education, occupational class, and income (Crude column). In our cohort, women had an approximately 33% lower risk of death than men (calculated from point estimate −0.402).

We then fitted a series of models adding the independent variables in the temporal order that they were measured or established, starting with parental income and ending with adult income. In Model 1, we adjusted for childhood cognitive ability, which attenuated the association between parental income and mortality (−0.001, *p* = 0.221). Higher cognitive ability was associated with lower mortality across all model specifications. Adding childhood social skills to the model did not substantially change the association between parental income and mortality (Model 2).

Next, we added indicators of adult socioeconomic status (Models 3–5). We did not observe an independent association between parental income and adult mortality in any of these models. Educational attainment had a graded association with mortality in that cohort members’ mortality risk decreased with increasing educational level. Including adult occupational class to the previous model did not alter the estimates of parental income, childhood cognitive ability, or social skills considerably, but it attenuated the estimates for educational attainment somewhat (Model 4). Finally, when adult income was included (Model 5), the associations of childhood cognitive ability, social skills, adult occupational class, and educational attainment with mortality were partly attenuated. Higher adult income was associated with lower mortality. In the same model, cohort members who had zero income had higher mortality but the estimate was not significant (coefficient is not presented in the table).

Income mobility patterns were then assessed by modeling tertiles of childhood parental income and adult income, and including interaction terms for all possible combinations of the two. The results of the interaction are presented in Figure 1 (the coefficients are presented in Table A1). We did not find a significant interaction between childhood parental income and adult income. However, there was a tendency towards smaller differences in mortality by adult income among individuals who had high-income parents during childhood. When we conducted a stratified analysis within the adult income tertiles and calculated the association with mortality by parental income tertiles, we saw a similar pattern (Table A2).

To test the robustness of our findings, we did two sets of sensitivity analyses. First, we re-introduced individuals back to our analytical sample that had previously been excluded due to missing information in one or several of the variables. The majority of these observations lacked information on either cognitive ability or social skills (n = 2236). Cohort members that were excluded because of missing information had a somewhat lower mean income (195,787 SEK) and a higher mortality (14.1%) compared to the analytical sample. We re-estimated models excluding cognitive ability and social skills in order to include more individuals (n = 13,479) and found similar results as those that were observed in the analytical sample (Table A3). Further, we found a significant mortality difference among low-income cohort members whose parental income was in the highest compared to the lowest tertile during the cohort members’ upbringing (Table A4 and Figure A1). This pattern was similar in the analytical sample, although not statistically significant.

Individuals with zero labor income are likely qualitatively different from other low-income earners. In order to account for this difference, we included a dummy variable indicating whether the individual had zero reported labor income. As our second sensitivity analysis, we excluded this group from our sample and re-estimated our models and found very similar results to our main results (Table A5 and Table A6, Figure A2).

## 4. Discussion

The aim of this study was to explore pathways between childhood economic conditions and adult mortality by taking personal attributes as well as adult socioeconomic career into consideration in the analyses. Lower childhood parental income was associated with higher mortality risk between ages 48–68, but this association was relatively modest with a one percentile-rank increase in parental income being associated with 0.004% lower mortality. Due to the attenuation of this association when we controlled for intermediate factors, in particular cognitive ability and achieved education, we conclude that the association operates through intermediate factors. This finding is consistent with the pathway model of social inequalities in health. We did not find a significant association between intergenerational income mobility and adult mortality.

Studies on the relationship between adult mortality and socioeconomic position in childhood and adulthood have typically considered how childhood and adult socioeconomic positions form pathways to disease and mortality independently or in combination with each other [1,2,7,12]. We extend this literature by considering intermediate factors between these two life stages, namely cognitive ability and social skills in adolescence. Our findings indicate that personal attributes play an important role in the pathway between childhood economic conditions, adult socioeconomic position, and adult mortality. We found independent associations between personal attributes and mortality after adjusting for adult education, occupation, and income. This implies that personal attributes can potentially influence adult mortality in two ways. First, it can matter for an individual’s attained socioeconomic position, which in turn affects mortality. Secondly, it can affect mortality independently of socioeconomic position.

These findings are consistent with previous research showing that personal attributes account for part of the socioeconomic health disparities [20,41,42] and are independently related to mortality [22]. The remaining association between personal attributes and mortality after adjustments has previously been explained through at least three mechanisms. Cognitive ability and other personal attributes can lead to better health-related behaviors (e.g., smoking, diet, physical exercise, etc.) [23]. They might also reflect genetic factors influencing both personal attributes and mortality [43]. At the same time, omitted childhood socioeconomic factors might also confound this relationship [20].

Also with regard to the role of adult socioeconomic career for mortality risk, our results are in line with previous findings [5,7,9,12,13]. Our results indicate that these findings are robust when taking cognitive ability and social skills into consideration in the analyses. Moreover, the results of this study indicate that childhood socioeconomic position influences mortality risk partly through adult socioeconomic position. Prior research has indicated that out of education, occupation, and income, educational attainment is the most important channel through which parental income affects adult mortality [2,7]. Educational attainment is usually completed before or during young adulthood and is determined by both personal attributes and childhood socioeconomic conditions [44]. Uniquely, we take this line of research further and show that cognitive ability, measured before adult socioeconomic characteristics are obtained, can be an important intermediate factor between parental income and mortality. Contrary to expectations, social skills played no such intermediate role, even though it was independently associated with mortality.

We found stronger associations between adult income and mortality compared to parental income during childhood [9,12,13]. This can be interpreted as adult income being more important for later mortality than childhood income. However, because of the bidirectional relationship between adult income and health, it is difficult to compare the estimates for childhood and adult income on mortality. Unlike childhood income, adult income is, in part, dependent on adult health status. Adult income is furthermore measured in closer proximity to the outcome compared to childhood income. A long period between exposure and outcome introduces more random variation which tends to bias associations towards the null [45]. It is also important to keep in mind that childhood conditions influence socioeconomic trajectories and is associated with mortality in part through adult income.

We assessed income mobility through the interaction between parental and adult income. While an interaction model is not a perfect way to distinguish the effects of origin and destination from the movement in itself [46], other statistical methods, such as the diagonal reference model [46], raise concerns as well [47]. However, our approach allows us to examine associations between parental income and mortality at different parts of the income distribution in adulthood. We did not find a significant interaction between parental and adult income, but an interesting tentative pattern nevertheless emerged from our data. Cohort members with low-income parents during childhood who reached a high adult income had the same mortality risk as cohort members with high-income parents and high adult income. Individuals who moved down in the income distribution compared to their parents, had a tendency towards lower mortality risk compared to those with low income in both childhood and adulthood. These results for downward mobility are consistent with most of the social mobility literature to date [6,25,27]. However, we did not find any indication that were in line with the recent findings suggesting that upward mobility may have a detrimental effect on health [28,29,30].

### Methodological and Historical Considerations

Our data material consists of high-quality register information on parental income, personal attributes and adult socioeconomic career that was not subject to recall bias. To address attenuation bias, we combined two years of income from both parents. Nevertheless, including only two years of parental income in a way that allows a large dispersion of parental age might lead to downward biased estimates of parental income on adult mortality. Further, the bidirectional nature of the relationship between adult income and health makes it difficult to draw conclusions about the processes that generate the observed association. The relationship between parental income and adult mortality, on the other hand, is not bidirectional. Moreover, sociometric popularity might be a weak proxy of social skills, which could be the reason why this personal attribute did not seem to play an intermediate role in the relationship between parental income and adult mortality in our analyses. Finally, due to the possibility of unobserved confounding, for example from genetic and environmental factors, the estimates that were presented in this study should not be interpreted as causal effects.

Even though our study aimed to explore pathways between childhood economic conditions and mortality, we decided not to use more advanced mediation analysis or structural equation models. These techniques typically rely on a series of assumptions on the causal relationships between the variables that are included in the analysis or require panel data with repeated measures of exposures and mediators. We refrained from using such methods and opted not to decompose the associations into direct and indirect effects of childhood factors because we are not confident in assuming causal relationships between the included variables. Moreover, to avoid the false implication of causal relationships, we applied the term ‘intermediate’ instead of mediating variables.

We followed the cohort members for mortality until May 2021 when they were 68 years old. Only 9.2% of the cohort members died during this period, and it is possible that a longer follow-up, enabling us to observe more deaths, would have led to different patterns. For example, a study by Sandberg and Palme [10] that followed an older Swedish cohort that was born in Malmö in 1928, found an association between parental income and mortality after adjusting for adult socioeconomic position. However, that study did not use information on personal attributes [10]. Typically, socioeconomic inequalities in mortality decline with age, in particular when mortality is measured on a relative scale.

Our cohort consists of children who were raised in an opportune time. The cohort members had plenty of chances to achieve upward mobility since they grew up during a time of expansion of the welfare state and declining inequalities [48,49,50]. A comprehensive school system reform took place while the cohort members attended school, expanding compulsory schooling to nine years [50], large scale housing programs throughout Stockholm improved general housing conditions [48], and income inequalities started to decline sharply in the 1960s due to welfare state policies aiming to reach economic security for everybody [49]. Under such societal conditions, personal attributes might become more salient for an individual’s opportunities for achieving high income and intergenerational income mobility. Thus, economic conditions in childhood might not have been as important as in previous and later generations, something which could serve as a fruitful topic for future studies.

## 5. Conclusions

In a cohort that experienced declining social inequalities and growing opportunities for upward mobility, parental income during childhood was associated with adult mortality mainly through cognitive ability and adult educational attainment. It is important to note that this does not mean that childhood economic conditions are not important for later-life mortality but, in our cohort, these effects were modest and mainly operated through the pathways that were related to cognitive ability and educational achievement leading up to mortality.

## Figures and Tables

**Figure 1 ijerph-19-07279-f001:**
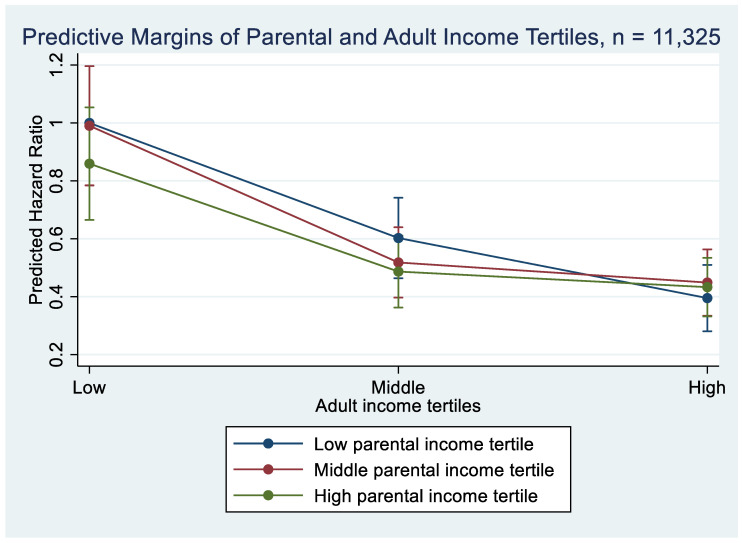
Predictive margins of adult income tertiles by parental income tertiles. Hazard ratios calculated based on Model 6 in Table A1.

**Table 1 ijerph-19-07279-t001:** Descriptive statistics of the studied cohort (born in 1953 and residing in Stockholm, Sweden, in 1963). Mortality follow-up 2001–2021 (n = 11,325).

Childhood characteristics	Mean	Std. Dev.
Parental income ^a^ (age 10, 17)	170,450.69	97,564.72
Cognitive ability (range: 12–116)	69.33	17.36
Social skills ^b^ (range: 0–12)	2.64	1.75
Adult characteristics	Frequency	Percentage
Educational attainment (age 37)		
Pre-primary	52	0.46
Primary	1885	16.64
Lower secondary	3363	29.70
(Upper) secondary	1638	14.46
Post-secondary non-tertiary	1982	17.50
First stage of tertiary	2280	20.13
Second stage of tertiary	125	1.10
Occupational class (age 38)		
Semi- or unskilled manuals	1440	12.72
Skilled manuals	1223	10.80
Lower non-manuals	4393	38.79
Upper non-manuals	1904	16.81
Self-employed	504	4.45
Not known	1861	16.43
	Mean	Std. Dev.
Adult income ^a^ (age 37–48)	225,484.73	149,030.11
Sex	Frequency	Percentage
Male	5360	47.33
Female	5965	52.67
Deceased between 2002–2021 (age 49–68)	1046	9.24

^a^ Income measures expressed in 2001 SEK; ^b^ Measured by sociometric popularity (total received friendship nominations).

**Table 2 ijerph-19-07279-t002:** Associations between all-cause mortality and parental income, childhood cognitive ability, and social skills as well as attained education, social class, own adult income, and sex. Cox regression, mortality follow-up 2001–2021. Individuals born in 1953 and residing in Stockholm, Sweden in 1963. (n = 11,325).

	Crude	Model 1	Model 2	Model 3	Model 4	Model 5
Parental income rank (age 10, 17)	−0.004 ***	−0.001	−0.001	0.001	0.001	0.002
Cognitive ability (age 13)	−0.015 ***	−0.015 ***	−0.014 ***	−0.007 ***	−0.007 ***	−0.005 **
Social skills ^a^ (age 13)	−0.080 ***		−0.059 **	−0.050 **	−0.046 *	−0.038 *
Educational attainment (age 37):						
Pre-primary	0.357			0.272	0.195	0.070
Primary (ref)	0			0	0	0
Lower secondary	−0.315 ***			−0.290 ***	−0.228 **	−0.198 *
(Upper) secondary	−0.564 ***			−0.466 ***	−0.418 ***	−0.340 **
Post-secondary non-tertiary	−0.890 ***			−0.784 ***	−0.720 ***	−0.613 ***
First stage of tertiary	−1.033 ***			−0.895 ***	−0.841 ***	−0.688 ***
Second stage of tertiary	−1.696 ***			−1.504 **	−1.501 **	−1.215 *
Occupational class (age 38)						
Semi- or unskilled manuals (ref.)	0				0	0
Skilled manuals	−0.511 ***				−0.486 ***	−0.453 ***
Lower non-manuals	−0.506 ***				−0.207 *	−0.131
Upper non-manuals	−0.745 ***				−0.136	−0.009
Self-employed	−0.271				−0.087	−0.205
Not known	0.091				0.243 *	−0.259 *
Log adult income ^b^ (age 37–48)	−0.430 ***					−0.375 ***
Sex						
Male (ref.)	0	0	0	0	0	0
Female	−0.402 ***	−0.470 ***	−0.483 ***	−0.439 ***	−0.447 ***	−0.563 ***
Log likelihood	−9681.064 ^c^	−9648.351	−9643.153	−9599.734	−9575.310	−9500.827

^a^ Measured by sociometric popularity (total received friendship nominations); ^b^ Models including log adult income are also adjusted for a binary variable indicating zero income values; ^c^ Log likelihood presented for the Crude model including parental income and sex; * *p* < 0.05, ** *p* < 0.01, *** *p* < 0.001.

## Data Availability

As the data consist of sensitive personal information, the data cannot be made publicly available. Any data-related inquires can be emailed to the corresponding author.

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
