# Peer review of "Pathways from Childhood Economic Conditions to Adult Mortality in a 1953 Stockholm Cohort: The Intermediate Role of Personal Attributes and Socioeconomic Career"

_ijerph, 2022, doi:10.3390/ijerph19127279_

Round 1

Reviewer 1 Report

Overall, my opinion is that this manuscript is very interesting and appropriate for this Journal's readers. The manuscript is very well written. I have only two suggestions that might improve it:

1.- Check references and reference style.

2.- “You do not really understand something unless you can explain it to your grandmother.” This reviewer suggests adding a plain-language summary or easy-to-read bullet data in brief.

Author Response

Point 1: Check references and reference style!

Response 1: We have checked and edited the references to confirm with the style.

Point 2: “You do not really understand something unless you can explain it to your grandmother.” This reviewer suggests adding a plain-language summary or easy-to-read bullet data in brief.

Response 2: Thank you for the suggestions. We added two paragraphs to the introduction (line 94); one summarizes the research gap and clarifies our contribution to the literature and the other describes our data and expected results.

Reviewer 2 Report

This study examines the role of intermediate factors between childhood income and adult mortality. This is a fantastically challenging task that only theoretically trained epidemiologists can tackle. Moreover, such a task can only be solved by having powerful databases which appear to be operational in Sweden. The rationale for the study is well described in the article. Statistical analysis was based on a prospective cohort study of individuals born in 1953 and residing in Stockholm, Sweden in 1963, who were followed for mortality between 2002 and 2021, using Cox proportional hazards models to assess the association of parental income, cognitive ability, social skills, educational attainment, occupational status, and adult income with mortality. The main conclusion was that  childhood economic conditions were found to play a distinct but relatively small role for later mortality

The article is written in a typical format. I did not notice any significant shortcomings with regard to the material presented in the article. Nevertheless, while reading the submitted manuscript, several questions arose and inaccuracies were noticed, which I recommend to fix not only personally for the reviewer but probably to readers too.

  1. Methods: Please argue why the Cox model rather than structural equation modelling was better at solving this problem.
  2. Is it possible to differentiate what was direct and indirect effect of childhood factors on adult mortality?
  3. Table 1: provide information about sex (numbers and coding).
  4. Table 2. What was the impact of sex (relation of mortality between sexes)?
  5. Appendix: In titles of the tables indicate what subjects were excluded from the analysis.
  6. Discussion: The strengths and limitations of the study should be emphasized more.

Thank you for considering my opinion. I encourage authors to keep on working to improve the manuscript.

Author Response

Thank you for your valuable comments! Please see the attachment.

Reviewer 3 Report

Thank you for your paper, I do believe it will make a valuable contribution to the discussion concerning the human capital and help us to understand the relationship between childhood and adult economic conditions. Your paper is well written, and interesting. However I do suggest some changes and points to address below:

Firstly, I think the author should tell the readers that how to choose the sample. And the author should add the province distribution, household characteristic of the sample and other information. For example gender, age and so on. Because of the difference behavior of age stage, the author should give us more information of the age instead of the mean of the age. For example the number of the young people, the number of the old people.

Secondly, the analysis and test is not enough. The author should use the Mechanism method to test the different effect between the male and female.

Author Response

(The authors gave the same response as above.)

Reviewer 4 Report

The manuscript contains research aimed at studying the effects of childhood economic conditions and intergenerational income mobility on adult mortality.

It is a well-written and structured work. I will only indicate some considerations that the authors may consider, if they deem it convenient, to improve the presentation of the work.

  • The Introduction should indicate what the gap covered by this work is, i.e., what it really contributes to the literature. It is true that the authors describe it, but it is not clearly established. This should be done in a separate paragraph summarizing the contribution.
  • At the end of the introduction, the authors could add a paragraph explaining the structure of the manuscript, as usual.
  • The authors could introduce a literature review section, separate from the introduction, where the research hypotheses are stated. In this way, the reader would know what to expect from the results. The Introduction should be reduced and part of its content should be moved to the Literature Review, which should be completed with more references.
  • The limitations of the study should be clearly stated at the end of the Discussion section.

Author Response

(The authors gave the same response as above.)

Reviewer 5 Report

I do not understand the connection between these people living in Sweden in 1963, did the number of respondens n=11.325 refer to these people?

Author Response

(The authors gave the same response as above.)

Round 2

Reviewer 3 Report

no other comments